# Industrial Organic Wastewater through Drip Irrigation to Reduce Chemical Fertilizer Input and Increase Use Efficiency by Promoting N and P Absorption of Cotton in Arid Areas

Xianzhe Hao [1,2,3,†], Xiaojuan Shi [1,†], Aziz Khan [1], Nannan Li [1], Feng Shi [1], Junhong Li [1], Yu Tian [1], Peng Han [1], Jun Wang [2,3,*] and Honghai Luo [1,*]

[1] Key Laboratory of Oasis Eco-Agriculture, Xinjiang Production and Construction Group, Shihezi University, Shihezi 832000, China
[2] Soil and Water Research Institute, Xinjiang Academy Agricultural and Reclamation Science, Shihezi 832000, China
[3] Key Laboratory of Northwest Oasis Water-Saving Agriculture, Ministry of Agriculture and Rural Affairs, Shihezi 832000, China
\* Correspondence: wjlp86100@163.com (J.W.); luohonghai@shzu.edu.cn (H.L.)
† These authors contributed equally to this article.

**Abstract:** The use of industrial waste as an agricultural resource is important for clean and sustainable agriculture. We assumed that industrial organic wastewater coupled with chemical fertilizer would increase cotton yield by enhancing nutrients absorption and utilization. To test this hypothesis, a two-year (2019–2020) field trial was conducted to assess the impacts of CK (0 kg ha$^{-1}$), chemical fertilizer (CF) (N-P2O5-K2O: 228-131-95 kg ha$^{-1}$), chemical fertilizer + organic wastewater (F0.6 (60%CF + OW: 1329 kg ha$^{-1}$), F0.8 (80%CF + OW), F1.0 (CF + OW), F1.2 (120%CF + OW) and F1.4 (140%CF + OW)) on nutrient absorption and distribution, fertilizer use efficiency and cotton yield under drip irrigation system. Compared with CF, the soil organic matter, NH4+-N and AV-K increased significantly after F0.8-F1.4 treatments. The absorption of nitrogen (N), phosphorus (P) and potassium (K) by plants after dripping organic wastewater (F0.8-F1.4) increased by 1.1–11.2% as compared with CF (F0.6, CF < F0.8, F1.0 < F1.2, F1.4). Under F0.8, treatment resulted in a higher distribution rate of N, P and K in reproductive organs compared with other counterparts. In addition, drip application of organic wastewater promoted the absorption of magnesium (Mg) and zinc (Zn) in leaves and Fe in roots with higher translocation of Zn and boron (B) to reproductive organs compared with other treatments. The absorption of N, P and K was positively correlated with Mg, negatively correlated with calcium (Ca) and sulfur (S), and positively correlated with manganese (Mn) and iron (Fe). The yield and fertilizer utilization rate of cotton were higher at F0.8. Conclusively, the use of 1329 kg ha$^{-1}$ organic wastewater (organic mattered ≥ 20%, humic acid ≥ 20 g L$^{-1}$, *Bacillus subtilis* ≥ 2 × 108 L$^{-1}$) combined with chemical fertilizer (N-P2O5-K2O) at (182-104-76 kg ha$^{-1}$) reduces the application of chemical fertilizer and can increase utilization efficiency of chemical fertilizer with a high cotton yield under mulch drip irrigation in arid regions.

**Keywords:** organic wastewater; fertilizer utilization rate; nutrient absorption; microelement





## 1. Introduction

Cotton (*Gossypium hirsutum* L.) is one of the most important natural fiber crops [1]. China is the second largest cotton producer and consumer worldwide with a total cotton output accounting for 25.6% of the global market in 2020, of which 87.3% is contributed by Xinjiang [2]. With climate warming and the improvement of intensive cropping systems, Xinjiang has become an important province with high cotton production [3,4]. The continuous use of chemical fertilizer has boosted crop yield [5]; the continuous growth of cotton production in Xinjiang could not have been realized without the extensive use of chemical fertilizers, pesticides and plastic films [6].

Continuous cropping for several years and the pursuit of high yields have resulted in high water and fertilizer input for cotton production [7]. In Xinjiang, the application amount of chemical fertilizer per unit area for cotton has been maintained at about 585 kg ha$^{-1}$ (N), but high fertilization intensity has not increased yield per unit yield [8], and has been detrimental to the environment such as causing groundwater pollution, soil degradation and greenhouse gas emissions [9–11]. Therefore, the development of efficient and sustainable cotton production management strategies is needed to achieve high cotton yield with minimal damage to the environment [12].

Drip irrigation systems are considered as efficient technology under severe water shortage, particularly in arid areas [13,14]. The application of drip irrigation in combination with plastic film not only conserves water resources, but can also increase cotton yield in dry regions [6,15]. The fertilization method in cotton fields has also changed from the traditional "base fertilizer plus top dressing" to dripping fertilization [16]. However, this fertilization mode of integration of water and fertilizer relies heavily on the use of chemical fertilizers resulting in less or no use of organic fertilizers [17]. Long-term use of chemical fertilizers can offset cultivated land quality and reduces fertilizer utilization efficiency [18,19]. Organic fertilizer is the best substitute for chemical fertilizer and can alleviate environmental pollutions induced by chemical fertilizer as well as improves soil health with increased crop yield [20–22]. However, most of the studies have been focused on solid organic fertilizer such as manure and compost [23,24], and contain insufficient nutrients with long release time [25]. Prolonged use of such fertilizer will aggravate heavy metal pollution [26]. Therefore, it is necessary to explore the rational application of organic fertilizer with chemical fertilizer for crops and the development of eco-friendly and cost-efficient organic materials [27].

Proper utilization of industrial wastes can reduce the negative impact on the ecological environment [28]. Organic wastes can not only contain nitrogen (N), phosphorus (P) and potassium (K), but are also comprised of growth hormones, amino acids, organic matter and humic acid [29,30]. However, the use of organic wastewater alone is not enough for plant growth due its low concentration of N, P and K [31]. In addition, the nutritional requirements of field crops are often concentrated in a relatively short period of time [25]. Organic liquid fertilizer can be uniformly applied together with irrigation water [32]. Meanwhile, it can improve the physical and chemical properties of soil and enhance disease resistance of the crop [33] and adjust the structure of the rhizosphere microbial community to improve crop growth [34]. Our previous research indicates that, based on the application of conventional chemical fertilizers, dripping 1329 kg ha$^{-1}$ organic wastewater (organic matter $\geq$ 20%, humic acid $\geq$ 20 g L$^{-1}$, *Bacillus subtilis* $\geq$ 2 $\times$ 108 L$^{-1}$) helps to increase cotton yield [35]. However, no study has been conducted on how dripping organic wastewater coupled with chemical fertilizer will affect soil nutrient uptake and utilization efficiency by cotton plants to achieve optimal yield.

Therefore, we assumed that industrial organic wastewater coupled with chemical fertilizer would increase cotton yield by enhancing nutrient absorption and utilization. The objectives of this study were to assess: (1) how organic wastewater combined with chemical fertilizer improves cotton yield and nutrient use efficiency; (2) the relationship between nitrogen and phosphorus utilization efficiency and the absorption and distribution of plant nutrients; (3) the mechanism of how organic wastewater combined with chemical fertilizer can contribute to clean and sustainable cotton production.

## 2. Materials and Methods

### 2.1. Experimental Site Description

The experiment was conducted during 2019–2020 at Xinjiang Academy of Agricultural Reclamation station (45°38′ N, 86°09′ E, 442.9 m above sea level) (Figure 1). The meteorological data during cotton planting in 2019 and 2020 are shown in Figure 2. The total precipitation of both years was 189.7 mm and 64.8 mm, and the accumulated temperatures were 4164.9 °C and 4240.8 °C, respectively. The soil was sandy loam, and the basic

nutrients of the 0–20 cm soil layer in 2019 were as follows: organic matter 15.00 g kg$^{-1}$, alkali-hydrolyzable nitrogen 42.20 mg kg$^{-1}$, available phosphorus 19.81 mg kg$^{-1}$, available potassium 274.28 mg kg$^{-1}$, total calcium 51.11 g kg$^{-1}$, total magnesium 17.64 g kg$^{-1}$ and total sulfur 1.03 g kg$^{-1}$.

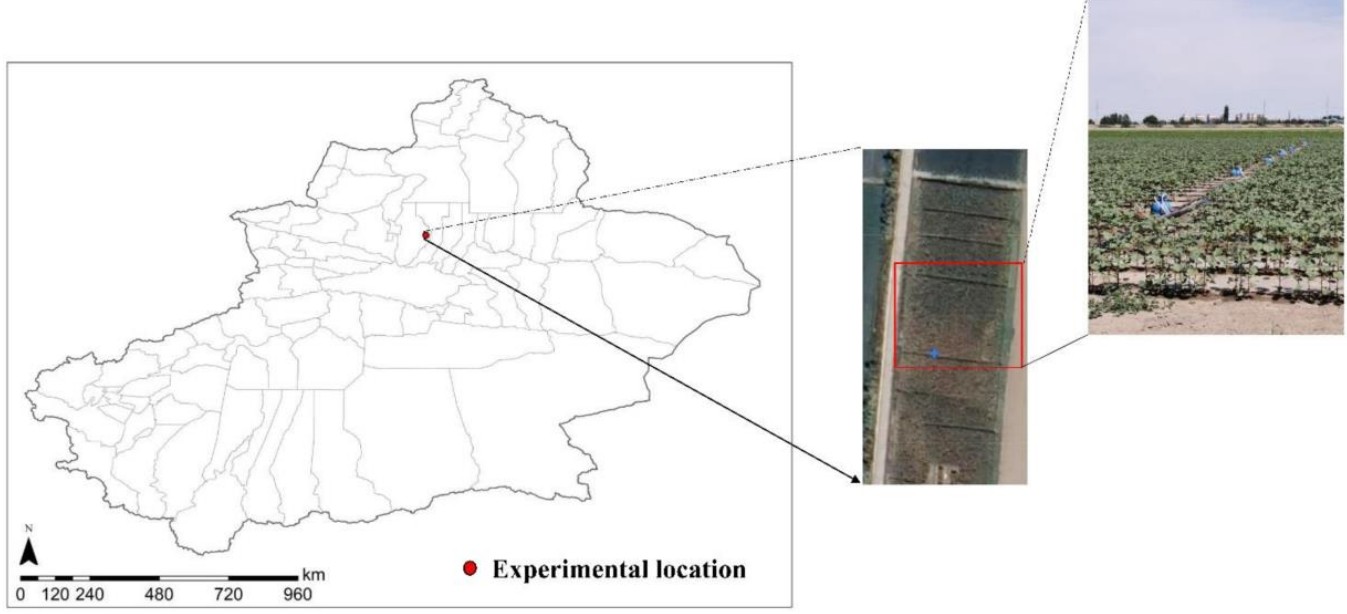

**Figure 1.** Location of the experimental area—Shihezi, Xinjiang, China.

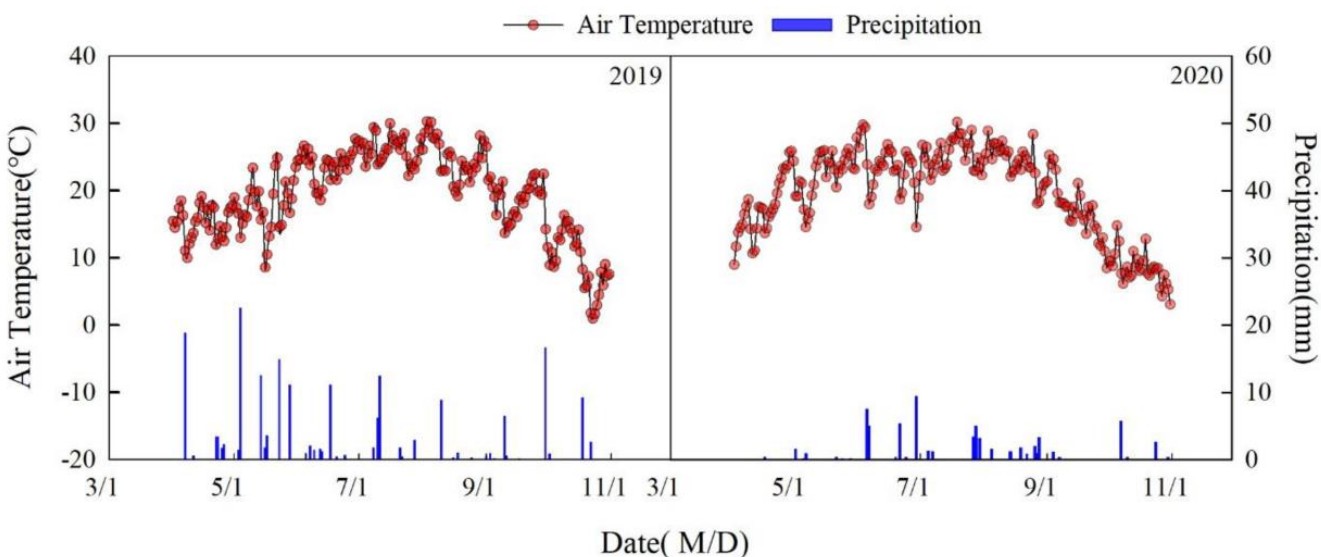

**Figure 2.** Average air temperature and precipitation in 2019 and 2020 growing period.

### 2.2. Experimental Design

A randomized complete block design was used. Seven treatments including no fertilization (CK), chemical fertilizer (CF) and organic wastewater combined with chemical fertilizer (F0.6, F0.8, F1.0, F1.2 and F1.4) were used with replications. In total, 21 plots with an area of 69 m$^2$ (6.9 m × 10 m) for each plot were used. The application amount of N, P$_2$O$_5$ and K$_2$O under F1 treatment was used according to the method in [36]. The amount of organic wastewater was adopted accordingly [35]. The application amount of N, P$_2$O$_5$ and K$_2$O under CF treatment was consistent with that of F1.0, while that under F0.6, F0.8, F1.2 and F1.4 treatments was 60%, 80%, 120% and 140% of that of F1.0 (Table 1).

Fertigation was carried out 9 times during the whole growth period. In brief, the details of this process are as follows. (1) Before each fertilization, water was dripped for about 0.5–1 h. (2) The fertilization ball valve was closed 50% for fertigation. (3) After 2–4 h of fertigation, the valve was fully opened and water was dripped until the required irrigation amount. Capacity differential pressure fertilization was adopted, and the fertilizer was stored in a 15 L plastic fertilizer tank. Other recommended management strategies were used for high-yield cotton fields under drip plastic mulching. The fertilizers such as urea (N content-46.0%), ammonium dihydrogen phosphate (N content-12.0% N and $P_2O_5$ content-61.0%) and potassium sulfate ($K_2O$ content-50.0%) were used. The organic waste used in this research is made of yeast wastewater produced by Angel Yeast (Yili) Co., Ltd. (Yili, China). as a carrier, adding *Bacillus subtilis*. The detail is given in Table 2.

*2.3. Data Collection*

2.3.1. Soil Properties

Seven days before sowing, a soil drill was used to collect basic soil samples according to the S shape method. During harvesting, soil was sampled (0–20 cm depth) from three different points in each plot. Samples were air-dried and sieved. Soil available phosphorus (AV-P) was determined by 0.5 mol $L^{-1}$ $NaHCO_3$ extraction–molybdenum antimony anti-colorimetry. The soil available potassium (AV-K) was determined by 1 mol $L^{-1}$ $NH_4OAc$ extraction–flame photometry. The soil organic matter was determined by the potassium dichromate–external heating method [37]. The contents of nitrate and ammonium nitrogen (AV-N) were extracted with 1 mol $L^{-1}$ potassium chloride solution and determined by automatic intermittent chemical analyzer (CleverChem380).

2.3.2. Cotton Nutrient Absorption

Six identical plants in each of the treatments were harvested at different growth stages such as the blooming stage, flowering stage, boll setting stage, late boll setting stage and boll opening stage. These samples were separated into different organs, including root (below cotyledon node), stem, leaf, bud and bell. Then, these samples were oven-dried at 105 °C for 30 min to deactivate enzymes. Further, the temperature was lowered to 80 °C to dried until a constant was obtained. The dried plant samples powered and stored for further analysis. Dried samples were digested with $H_2SO_4$-$H_2O_2$, to assess N uptake, P uptake and K uptake contents via Nessler colorimetry, vanadium molybdenum yellow colorimetry and flame photometry, respectively [38]. Nutrient accumulation in the overground part (mg $g^{-1}$) is equal to plant nutrient content multiplied by plant biomass.

2.3.3. Accumulation of Trace Elements in Cotton Plant

After crushing and sieving the samples in the flocculation stage, the samples were weighed to be measured. Then, $HNO_3$-HCl-HF was used to digest the acid system completely, and the Milestone Ethos microwave digestion instrument was used to heat and digest them and diluted with deionized water. Finally, calcium (Ca), magnesium (Mg), sulfur (S), iron (Fe), manganese (Mn), zinc (Zn), copper (Cu) and boron (B) concentrations in each treatment solution were quantitatively analyzed by ICP6300 [39].

2.3.4. Cotton Yield

In the harvest (22 September 2019 and 26 September 2020), three representative sampling sites (2.93 m $\times$ 2.3 m) were selected for each treatment. Number of plants per unit area and the number of bolls per plant in each sampling site were investigated. Meanwhile, 50 cotton bolls were picked from the top, middle and bottom of each plot (assuming the height of 1/3 as the boundary) to assess lint percentage and single boll weight.

**Table 1.** Fertilization schemes for different treatments.

| Treatment | Fertigation | Seeding | Squaring | | | Flowering and Boll Setting | | | | | | Total |
|---|---|---|---|---|---|---|---|---|---|---|---|---|
| | | 20-April | 15-Jun | 22-Jun | 1-Jul | 9-Jul | 18-Jul | 27-Jul | 5-Aug | 13-Aug | 20-Aug | |
| CF | Irrigation ($m^3 ha^{-1}$) | 300 | 525 | 300 | 375 | 450 | 600 | 600 | 600 | 300 | 300 | 4350 |
| | N ($kg ha^{-1}$) | 9.0 | 19.8 | 24.8 | 29.7 | 28.5 | 25.7 | 24.2 | 26.4 | 23.1 | 16.5 | 228.0 |
| | $P_2O_5$ ($kg ha^{-1}$) | 4.4 | 10.8 | 13.5 | 16.2 | 18.0 | 16.2 | 15.3 | 14.4 | 12.6 | 9.0 | 131.0 |
| | $K_2O$ ($kg ha^{-1}$) | 1.5 | 5.4 | 6.8 | 8.1 | 18.0 | 16.2 | 15.3 | 9.6 | 8.4 | 6.0 | 95.0 |
| | Irrigation ($m^3 ha^{-1}$) | 300 | 525 | 300 | 375 | 450 | 600 | 600 | 600 | 300 | 300 | 4350 |
| | Organic wastewater ($kg ha^{-1}$) | 46.9 | 122.0 | 141.1 | 169.3 | 179.9 | 161.9 | 152.9 | 145.7 | 127.5 | 91.0 | 1329 |
| F0.6 | N ($kg ha^{-1}$) | 5.4 | 11.9 | 14.9 | 17.8 | 17.1 | 15.4 | 14.5 | 15.8 | 13.9 | 9.9 | 136.8 |
| | $P_2O_5$ ($kg ha^{-1}$) | 2.7 | 6.5 | 8.1 | 9.7 | 10.8 | 9.7 | 9.2 | 8.6 | 7.6 | 5.4 | 78.6 |
| | $K_2O$ ($kg ha^{-1}$) | 0.9 | 3.2 | 4.1 | 4.9 | 10.8 | 9.7 | 9.2 | 5.8 | 5.0 | 3.6 | 57.0 |
| F0.8 | N ($kg ha^{-1}$) | 7.2 | 15.8 | 19.8 | 23.8 | 22.8 | 20.5 | 19.4 | 21.1 | 18.5 | 13.2 | 182.4 |
| | $P_2O_5$ ($kg ha^{-1}$) | 3.6 | 8.6 | 10.8 | 13.0 | 14.4 | 13.0 | 12.2 | 11.5 | 10.1 | 7.2 | 104.8 |
| | $K_2O$ ($kg ha^{-1}$) | 1.2 | 4.3 | 5.4 | 6.5 | 14.4 | 13.0 | 12.2 | 7.7 | 6.7 | 4.8 | 76.0 |
| F1.0 | N ($kg ha^{-1}$) | 9.0 | 19.8 | 24.8 | 29.7 | 28.5 | 25.7 | 24.2 | 26.4 | 23.1 | 16.5 | 228 |
| | $P_2O_5$ ($kg ha^{-1}$) | 4.4 | 10.8 | 13.5 | 16.2 | 18.0 | 16.2 | 15.3 | 14.4 | 12.6 | 9.0 | 131.0 |
| | $K_2O$ ($kg ha^{-1}$) | 1.5 | 5.4 | 6.8 | 8.1 | 18.0 | 16.2 | 15.3 | 9.6 | 8.4 | 6.0 | 95.0 |
| F1.2 | N ($kg ha^{-1}$) | 10.8 | 23.8 | 29.7 | 35.6 | 34.2 | 30.8 | 29.1 | 31.7 | 27.7 | 19.8 | 273.6 |
| | $P_2O_5$ ($kg ha^{-1}$) | 5.4 | 13.0 | 16.2 | 19.4 | 21.6 | 19.4 | 18.4 | 17.3 | 15.1 | 10.8 | 157.2 |
| | $K_2O$ ($kg ha^{-1}$) | 1.8 | 6.5 | 8.1 | 9.7 | 21.6 | 19.4 | 18.4 | 11.5 | 10.1 | 7.2 | 114.0 |
| F1.4 | N ($kg ha^{-1}$) | 12.6 | 27.7 | 34.7 | 41.6 | 39.9 | 35.9 | 33.9 | 37.0 | 32.3 | 23.1 | 319.2 |
| | $P_2O_5$ ($kg ha^{-1}$) | 6.3 | 15.1 | 18.9 | 22.7 | 25.2 | 22.7 | 21.4 | 20.2 | 17.6 | 12.6 | 183.4 |
| | $K_2O$ ($kg ha^{-1}$) | 2.1 | 7.6 | 9.5 | 11.3 | 25.2 | 22.7 | 21.4 | 13.4 | 11.8 | 8.4 | 133.0 |

**Table 2.** Properties of the organic wastewater used in this study.

| Organic Matter ($g L^{-1}$) | Humic Acid ($g L^{-1}$) | Macroelement ($g L^{-1}$) | | | Medium Trace Element ($g L^{-1}$) | | | | | | | pH | Liquid Densities ($g mL^{-1}$) | Microbial Flora *Bacillus subtilis* ($g L^{-1}$) | Heavy Metals (%) | | |
|---|---|---|---|---|---|---|---|---|---|---|---|---|---|---|---|---|---|
| | | N | P | K | Mo | B | Cu | Mn | Fe | Zn | Ca | | | | Pb | Cd | Cr |
| 208.1 | 22.1 | 23.4 | 1.7 | 56.6 | 0.08 | 0.2 | $6 \times 10^{-8}$ | 0.02 | 0.1 | 0.4 | 3.3 | 7.2 | 1.21 | $2 \times 10^{-8}$ | $2.5 \times 10^{-4}$ | $1.8 \times 10^{-5}$ | $4.7 \times 10^{-5}$ |

Abbreviations: molybdenum (Mo), boron (B), copper (Cu), manganese (Mn), iron (Fe), zinc (Zn), calcium (Ca), lead (Pb), cadmium (Cd), chromium (Cr).

2.3.5. Fertilizer Utilization Efficiency

The following formulas were used to assess different fertilizer utilization efficiencies [40]:

Nitrogen partial factor productivity (kg/kg) = yield/nitrogen application rate;

Nitrogen agronomic efficiency (kg/kg) = (yield in fertilization area-yield in no fertilization area)/nitrogen application rate;

Nitrogen use efficiency (%) = (nitrogen absorption in fertilization area-nitrogen absorption in no fertilization area)/nitrogen application rate × 100;

Phosphorus partial factor productivity (kg/kg) = yield/phosphorus application rate;

Phosphorus agronomic efficiency (kg/kg) = (yield in fertilization area-yield in no fertilization area)/amount of phosphorus applied;

Phosphorus use efficiency (%) = (phosphorus absorption in fertilization area-phosphorus absorption in no fertilization area)/phosphorus application amount × 100.

*2.4. Data Analysis*

Microsoft Office 2016 was used for data processing. SigmaPlot 14.0 (Systat Software Inc., San Jose, CA, USA) was used for drawing figures. SPSS19.0 (SPSS Inc., New York, NJ, USA) was used for data analysis, and one-way ANOVA and Duncan's method were adopted for multiple comparison. The significant differences among the treatment were separated at $p < 0.05$. In addition, Pearson correlations and Path analysis were performed to further assess the relationship between N/P/K and the other element using the "LinkET" "lavaan" package in R 4.0.3 [41].

**3. Results**

*3.1. Soil Nutrients*

Compared with CK, the CF treatment significantly increased the contents of available phosphorus, available potassium and ammonium nitrogen (Table 3). Organic wastewater significantly increased the contents of soil organic matter, available phosphorus, available potassium, nitrate nitrogen and ammonium nitrogen with an average increase of 22.5–806.3%. Compared with CF, organic wastewater application significantly increased soil organic matter, available phosphorus, available potassium, ammonium nitrogen and nitrate nitrogen (except available phosphorus of F0.6–F1.0 and nitrate nitrogen of F0.6) by 12.1–190.3%. There was no significant difference in organic matter, available phosphorus and available potassium among F0.6–F1.4 treatments.

**Table 3.** Effects of different fertilization patterns on soil nutrients in cotton field (2020).

| Treatment | Soil Organic Matter (g·kg$^{-1}$) | Available Phosphorus (mg·kg$^{-1}$) | Available Potassium (mg·kg$^{-1}$) | Nitrate Nitrogen (mg·kg$^{-1}$) | Ammonium Nitrogen (mg·kg$^{-1}$) |
|---|---|---|---|---|---|
| CK | 17.1 ± 0.7 b | 14.8 ± 0.9 c | 219.1 ± 9.0 c | 0.98 ± 0.09 d | 1.18 ± 0.02 c |
| CF | 17.7 ± 0.4 b | 21.0 ± 4.0 b | 239.5 ± 6.9 b | 3.06 ± 0.19 c | 1.26 ± 0.13 c |
| F0.6 | 22.8 ± 4.0 a | 24.1 ± 2.5 ab | 270.8 ± 4.2 a | 3.28 ± 0.13 c | 1.69 ± 0.14 b |
| F0.8 | 24.6 ± 4.7 a | 24.6 ± 3.2 ab | 269.6 ± 9.5 a | 8.53 ± 0.14 b | 1.88 ± 0.07 ab |
| F1.0 | 25.0 ± 3.6 a | 25.9 ± 2.1 ab | 262.5 ± 23.7 a | 8.65 ± 1.27 b | 1.83 ± 0.07 ab |
| F1.2 | 25.8 ± 3.4 a | 27.4 ± 8.4 a | 280.0 ± 26.4 a | 8.88 ± 0.71 b | 1.74 ± 0.07 b |
| F1.4 | 26.3 ± 3.0 a | 29.3 ± 8.7 a | 259.2 ± 19.8 a | 15.07 ± 0.02 a | 1.97 ± 0.07 a |

Note: Values are the means ± standard errors for three biological replicates. Different letters in the column represent significant differences ($p < 0.05$) based on Duncan's test.

*3.2. Absorption and Distribution of N, P and K*

Cotton plant N, P and K absorption showed an S-shaped curve (slow–fast–slow) (Figure 3). The absorption of N, P and K in cotton plants increased rapidly after 60 days of emergence in the following trend CK < CF, F0.6, F0.8 < F1.0, F1.2, F1.4. The absorption of N, P and K in fertilization treatments was significantly higher than that of CK at 120 days

after emergence ($p < 0.05$). After 120 days of cotton emergence, CF, F0.6 < F0.8, F1.0 < F1.2 and F1.4 in terms of N, P and K absorption for these fertilization treatments. There is no significant difference between F0.8 and F1.0 ($p < 0.05$). Compared with CF, dripping organic wastewater increased the absorption of N, P and K by 3.2–11.2%, 1.1–9.8% and 3.5–6.9%, respectively.

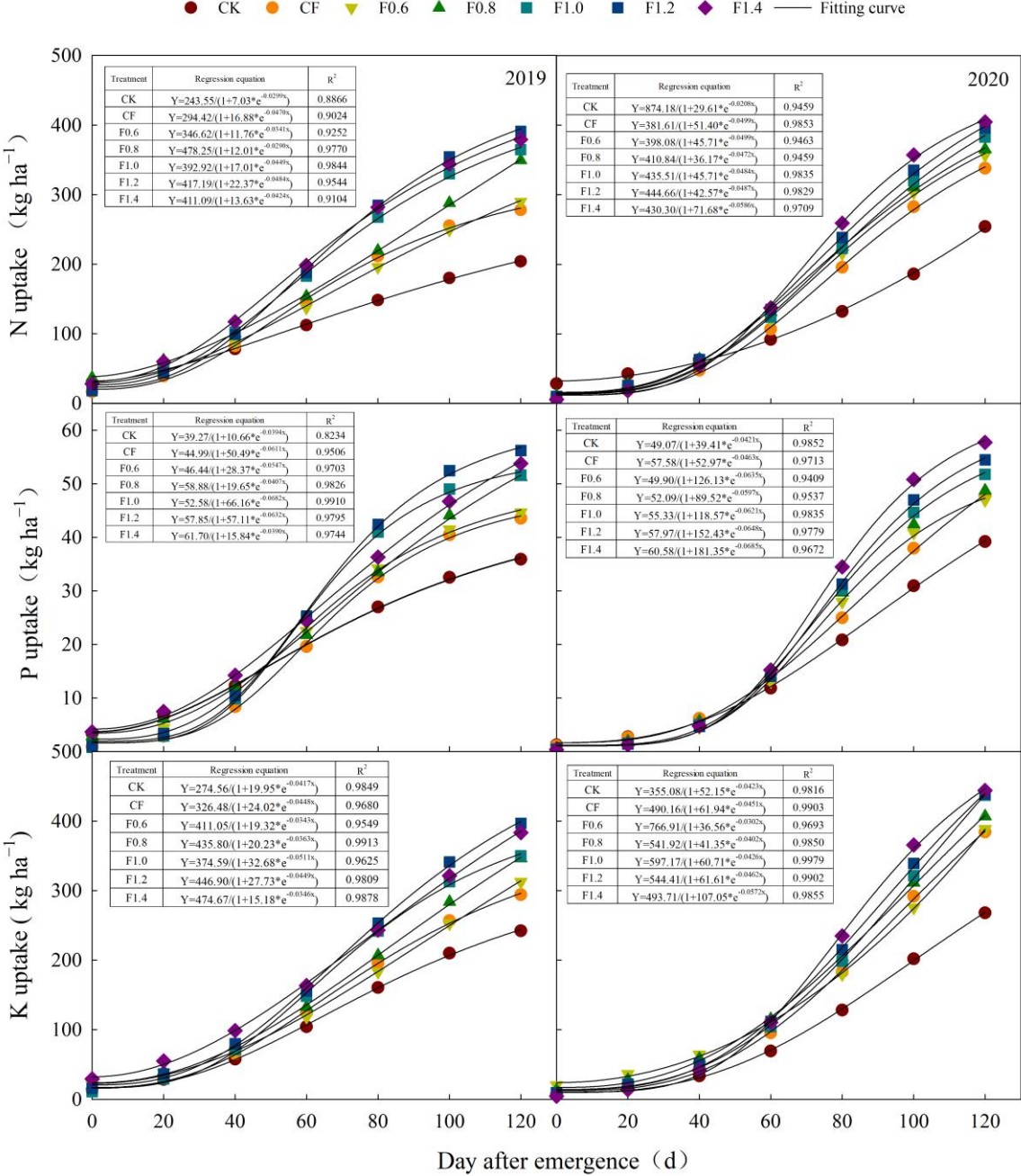

**Figure 3.** Changes in N, P and K accumulation in cotton plants under different fertilization in both years.

Compared with CF, the accumulation of N, P and K in reproductive organs under dripping organic wastewater was significantly increased ($p < 0.05$), and the average increases for F0.8–1.4 treatments were 14.5–0.8%, 2.3–19.7% and 13.3–20.7% (Figures 4–6). The F0.8 treatment had 73.8% and 83.6% higher distribution rates for N and P in reproductive organs. The highest distribution rate of K in reproductive organs was 65.6% under F1.2 treatment.

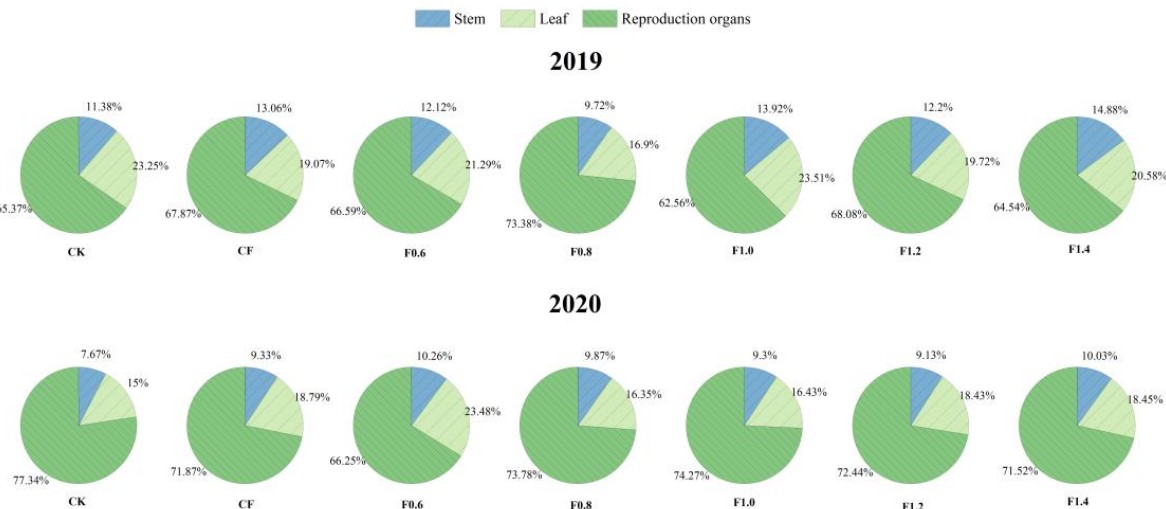

**Figure 4.** Distribution of N in different organs under different fertilization patterns during harvest.

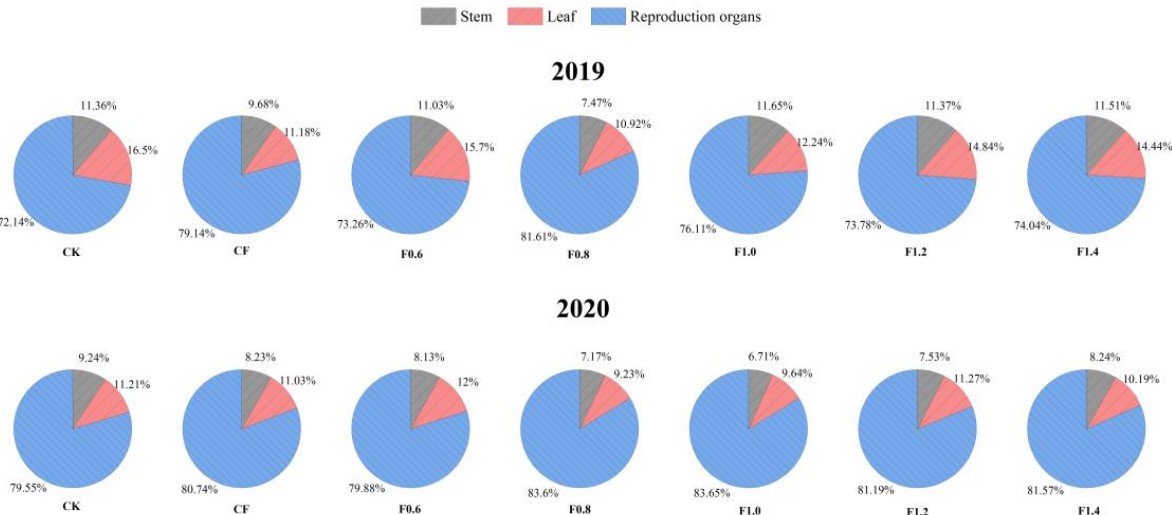

**Figure 5.** Distribution of P in different organs under different fertilization patterns during harvest.

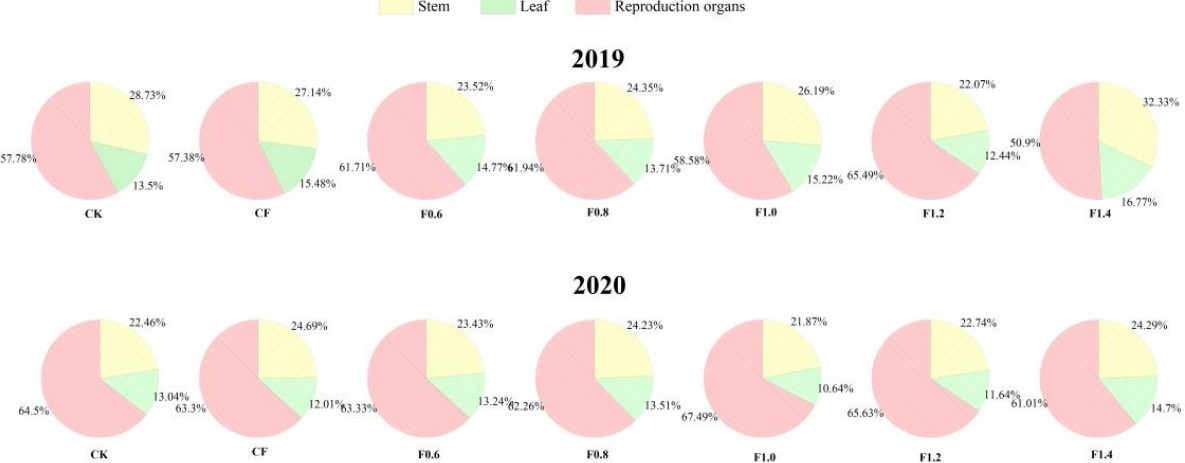

**Figure 6.** Distribution of K in different organs under different fertilization patterns during harvest.

### 3.3. Absorption and Distribution of Ca, Mg and S

Figure 7 shows that the absorption and accumulation of Ca was the highest in CK, and the absorption of Ca significantly decreased after dripping organic wastewater. The absorption of Mg in CF and F0.6–F1.4 was significantly higher than that of CK ($p < 0.05$), and the maximum increase was 8.2–16.4% in F0.8 treatment. The absorption of S decreased in the treatments with the dripping of organic wastewater (F0.6–F1.4), and F0.8 treatment had the largest decrease, which was 2.0–11.4% compared with that of CF.

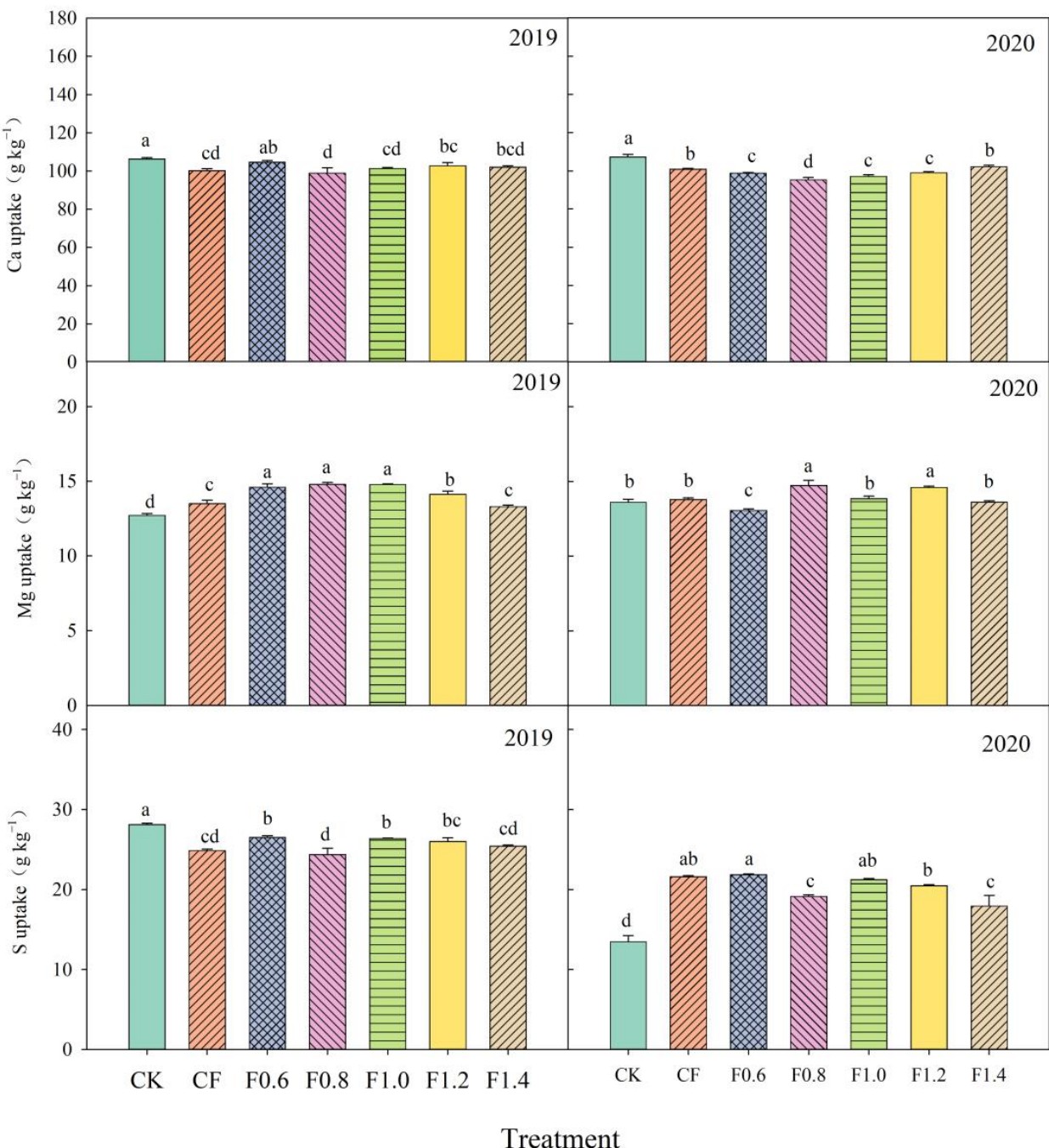

**Figure 7.** Change in Ca, Mg and S absorption in cotton under different fertilization patterns. Note: Values are means ± standard errors of three biological replicates. Bars with different letters in the same year are significantly different at $p < 0.05$.

After dripping organic wastewater, Ca distribution in stems decreased first and then increased with chemical fertilization (Figure 8). For Mg, compared with that of CK, the distribution rate of Mg in the leaves of fertilization treatment (CF, F0.8–F1.4) increased, while the distribution rate of bolls decreased (except F0.6). Compared with that of CF, dripping organic wastewater improved the distribution of Mg in reproductive organs. The distributions of the S element in different treatments were inconsistent in both years. For the application of organic wastewater, its distribution in leaves decreased, but increased in bolls.

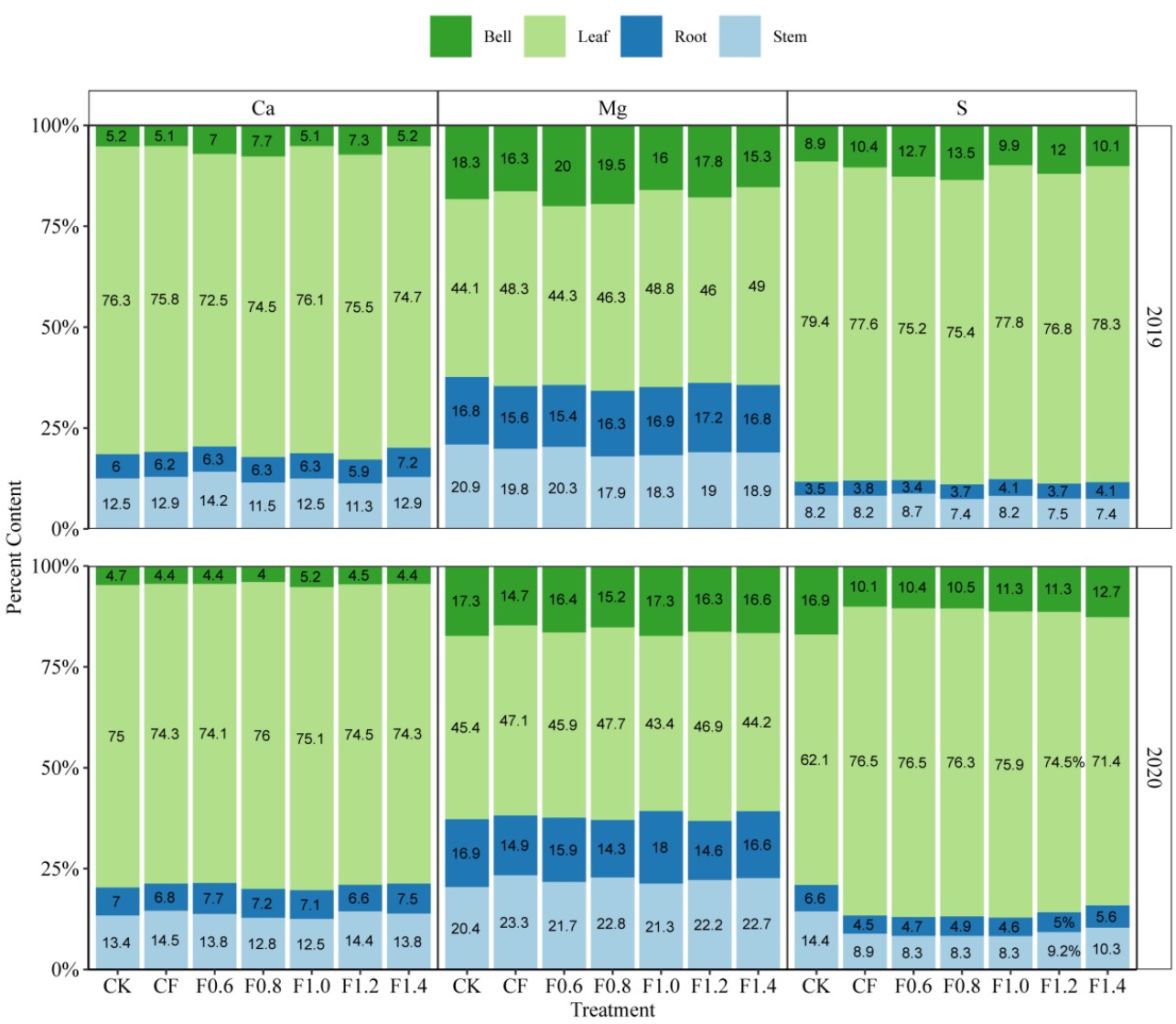

**Figure 8.** Changes in Ca, Mg and S distribution in cotton under different fertilization patterns.

*3.4. Absorption and Distribution of Cu, Zn, Mn, Fe and B*

The application of chemical and organic wastewater significantly impacted the absorption of Cu, Zn, Mn, Fe and B (Table 4). With the increase in chemical fertilizers, the absorption of Cu decreased. Under F0.6 treatment, Cu was 1.9–18.7% higher than that of CF on average. Compared with CF, the absorption of Zn under organic wastewater increased by 6.4–32.6%. Under F0.8 treatment, the absorption of Mn increased by 18.7–54.2% compared with CF ($p < 0.05$). Increasing dripped organic wastewater fertilization decreased Fe absorption, while under F0.8 Fe absorption, it increased by 14.4–25.0% compared with CF. Boron (B) absorption increased by 95.3% and 97.9% under F0.6 and F0.8, respectively, over CF. Compared with CF, dripping organic wastewater increased the distribution of

Cu, Zn and B in reproductive organs (Figure 9). F0.8 treatment resulted in 25.8–31.5%, 31.5–39.7% and 25.1–27% higher Cu, Zn and B accumulation, respectively, in reproductive organs. Compared with CF, fertilization treatment significantly increased the distribution of Mn in leaves and Fe in roots ($p < 0.05$).

**Table 4.** Changes in Cu, Zn, Mn, Fe and B absorption of cotton under different fertilization modes.

| Year | Treatment | Cu Uptake (mg kg$^{-1}$) | Zn Uptake (mg kg$^{-1}$) | Mn Uptake (mg kg$^{-1}$) | Fe Uptake (mg kg$^{-1}$) | B Uptake (mg kg$^{-1}$) |
|---|---|---|---|---|---|---|
| | CK | 12.9 ± 0.1 c | 42.9 ± 1.7 d | 98.5 ± 1.7 d | 1153.1 ± 9.9 d | 1.40 ± 0.04 f |
| | CF | 14.9 ± 0.3 bc | 44.7 ± 0.6 cd | 103.2 ± 1.4 c | 1174.5 ± 4.4 c | 1.75 ± 0.01 e |
| | F0.6 | 17.7 ± 0.9 a | 50.9 ± 2.1 ab | 103.1 ± 1.9 c | 1457.4 ± 7.6 a | 2.22 ± 0.07 c |
| 2019 | F0.8 | 16.8 ± 1.5 ab | 53.6 ± 2.1 a | 159.1 ± 1.1 a | 1467.9 ± 0.7 a | 2.56 ± 0.04 a |
| | F1.0 | 16.6 ± 0.6 ab | 48.2 ± 0.9 b | 154.6 ± 1.5 b | 1427.3 ± 9.7 b | 2.54 ± 0.00 a |
| | F1.2 | 16.7 ± 2.1 ab | 47.6 ± 1.5 bc | 153.2 ± 3.1 b | 1451.2 ± 16.5 a | 2.41 ± 0.02 b |
| | F1.4 | 13.3 ± 1.0 c | 44.0 ± 1.1 c | 152.9 ± 2.3 b | 1410.4 ± 8.2 b | 2.06 ± 0.03 d |
| | CK | 11.0 ± 0.4 b | 34.6 ± 3.3 c | 97.6 ± 2.6 c | 956.9 ± 7.7 e | 0.35 ± 0.00 f |
| | CF | 13.2 ± 0.2 a | 37.0 ± 1.0 c | 119.5 ± 0.4 b | 1331.8 ± 20.9 d | 0.43 ± 0.02 e |
| | F0.6 | 13.5 ± 0.4 a | 46.8 ± 1.8 a | 126.7 ± 0.2 b | 1451.3 ± 14.7 b | 1.15 ± 0.03 a |
| 2020 | F0.8 | 13.2 ± 0.2 a | 49.0 ± 1.0 a | 141.8 ± 2.1 a | 1523.6 ± 2.4 a | 1.08 ± 0.01 b |
| | F1.0 | 13.1 ± 0.3 a | 45.3 ± 1.0 ab | 126.1 ± 2.3 b | 1480.7 ± 12.4 b | 0.78 ± 0.01 c |
| | F1.2 | 13.1 ± 1.9 a | 44.7 ± 2.1 ab | 118.9 ± 1.7 b | 1379.7 ± 23.4 c | 0.66 ± 0.03 d |
| | F1.4 | 12.8 ± 1.0 ab | 41.7 ± 1.9 b | 116.5 ± 3.4 b | 1364.9 ± 7.6 cd | 0.41 ± 0.03 e |
| | Year | * | NS | ** | NS | ** |
| | Treatment | * | ** | ** | ** | ** |
| | Treatment × Year | NS | NS | ** | ** | ** |

Note: Values are the means ± standard errors for three biological replicates. Different letters in the column represent significant differences ($p < 0.05$) based on Duncan's test. * and ** represent $p < 0.05$ and $p < 0.01$, respectively, and NS represents "not significant".

*3.5. Cotton Yield and Fertilizer Utilization Rate*

Cotton yield was significantly impacted by both chemical and organic fertilizer in both years (Figure 10). Compared with that of CK, the yield of seed cotton after fertilization (CF, F0.6–F1.4) increased significantly ($p < 0.05$), approximately 4.4–44.6% (Figure 10). Compared with that of CF, the yield of seed cotton increased first and then decreased (F0.6–F1.4). The average increases for F0.6–F1.2 were 3.7–26.7% and F0.8 had the largest increase, 23.6% on average. It shows that the combination of organic wastewater with proper amounts of chemical fertilizers is more beneficial to the increase in cotton yield than that of applying chemical fertilizers alone.

The fertilizer utilization efficiency was significantly impacted by the application of chemical and organic fertilization in both years (Table 5). It can be seen that different treatments had the same NPEP and PPEP, and that of F0.6–F1.0 was significantly higher than CF ($p < 0.05$), with an average increase of 14.8–83.0%. For the change trend of NAE and PAE among treatments, F0.8 > F0.6 > F1.0 > F1.2 > CF > F1.4. The NAE and PAE of F0.8 increased by 36.6–37.1% compared with F0.6 ($p < 0.05$) and 47.3–60.6% compared with F1.4. For the NUE and PUE, that of F0.6–F1.4 increased by 47.5–221.2% and 23.3–296.9% on average compared with CF. Among them, F0.6 and F0.8 had the largest increase, 156.0–160.1% and 110.4–131.1% on average. The PUE had no significant difference among treatments ($p < 0.05$, 2020).

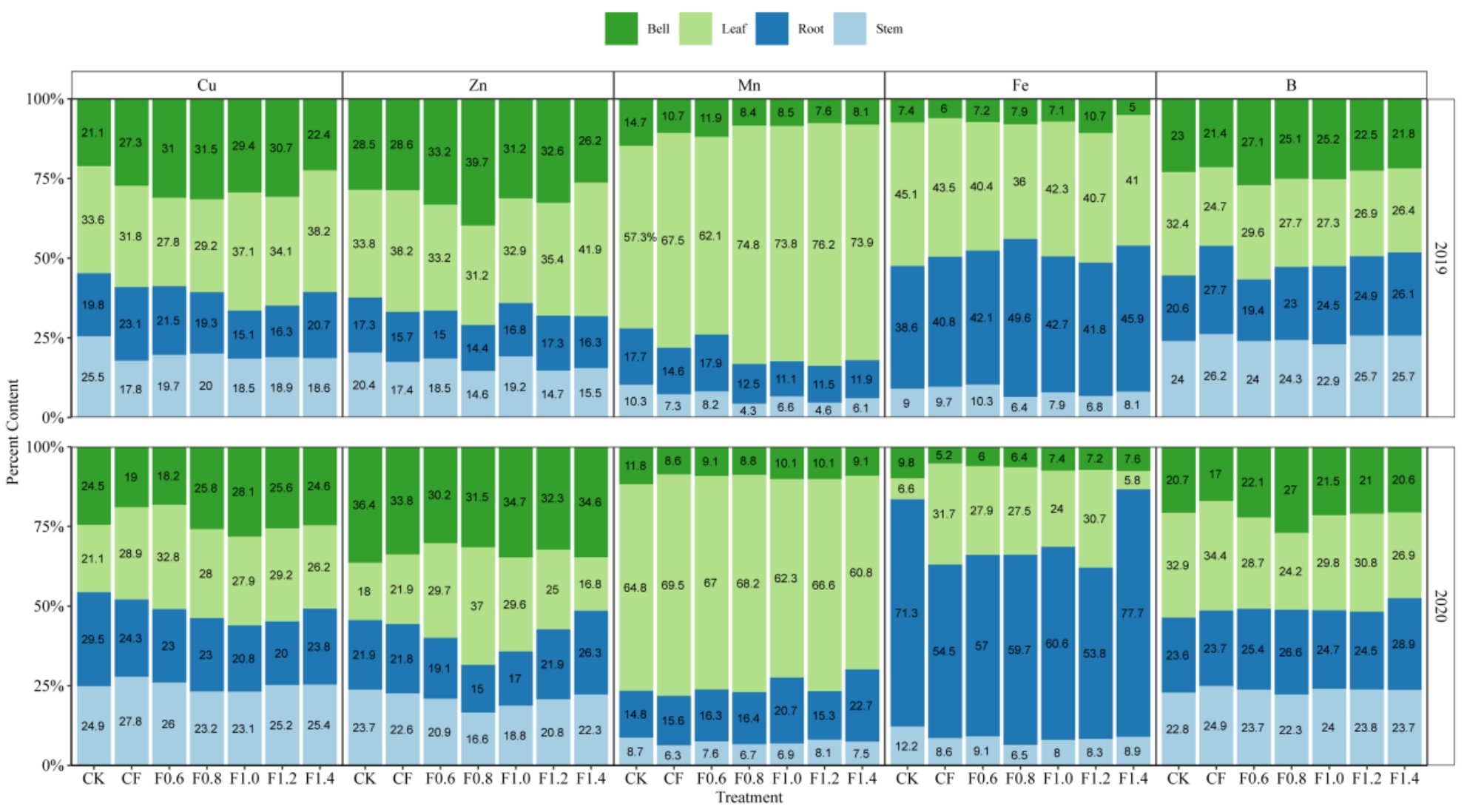

**Figure 9.** Changes in Cu, Zn, Mn, Fe and B distribution in cotton under different fertilization patterns.

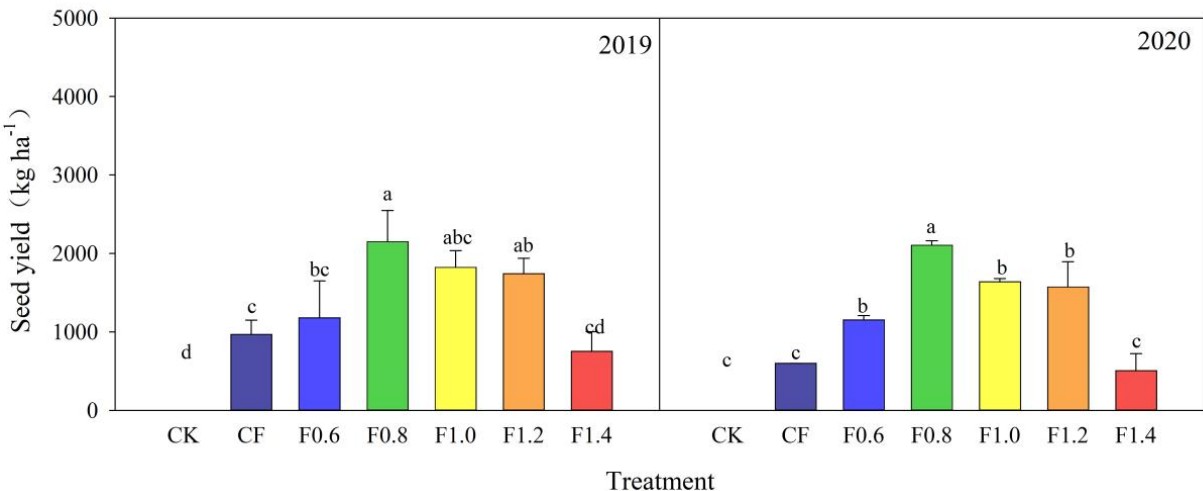

**Figure 10.** Difference in cotton yield between different fertilization treatments and CK. Note: Values are the means ± standard errors for three biological replicates. Bars with different letters in the same year are significantly different at $p < 0.05$.

**Table 5.** Effects of different fertilization modes on fertilizer utilization rate.

| Years | Treatment | NPEP (kg/kg) | NAE (kg/kg) | NUE (%) | PPEP (kg/kg) | PAE (kg/kg) | PUE (%) |
|---|---|---|---|---|---|---|---|
| 2019 | F0.6 | 43.98 ± 3.95 a | 8.64 ± 3.95 ab | 56.41 ± 5.60 c | 76.72 ± 6.88 a | 15.08 ± 6.88 ab | 9.11 ± 1.89 b |
| | F0.8 | 38.31 ± 2.58 b | 11.81 ± 2.58 a | 78.84 ± 11.74 a | 66.83 ± 4.49 b | 20.60 ± 4.49 a | 14.35 ± 2.16 a |
| | F1.0 | 29.22 ± 1.25 c | 8.02 ± 1.25 ab | 72.09 ± 3.65 ab | 50.97 ± 2.18 a | 13.99 ± 2.18 ab | 11.35 ± 1.82 ab |
| | F1.2 | 24.05 ± 0.98 d | 6.38 ± 0.98 bc | 63.27 ± 6.59 bc | 41.95 ± 1.70 d | 11.13 ± 1.70 bc | 12.17 ± 2.50 ab |
| | F1.4 | 17.51 ± 1.00 e | 2.37 ± 1.00 c | 48.60 ± 5.94 c | 30.54 ± 1.75 e | 4.13 ± 1.75 c | 9.32 ± 2.20 b |
| | CF | 25.45 ± 1.12 cd | 4.25 ± 1.12 bc | 24.54 ± 2.47 d | 44.39 ± 1.96 cd | 7.41 ± 1.96 bc | 3.61 ± 0.48 c |
| 2020 | F0.6 | 45.30 ± 0.96 a | 8.44 ± 0.96 b | 69.08 ± 6.28 a | 79.02 ± 1.68 a | 14.72 ± 1.68 b | 9.67 ± 1.43 a |
| | F0.8 | 39.22 ± 0.71 b | 11.57 ± 0.71 a | 55.99 ± 10.67 a | 68.41 ± 1.23 b | 20.18 ± 1.23 a | 8.45 ± 4.86 a |
| | F1.0 | 29.32 ± 0.64 c | 7.20 ± 0.64 b | 50.51 ± 2.65 b | 51.15 ± 1.12 c | 12.57 ± 1.12 b | 8.78 ± 0.65 a |
| | F1.2 | 24.19 ± 1.87 d | 5.76 ± 1.87 b | 47.86 ± 1.14 b | 42.19 ± 3.26 d | 10.04 ± 3.26 b | 8.61 ± 1.80 a |
| | F1.4 | 17.39 ± 1.26 e | 1.59 ± 0.14 c | 43.30 ± 1.09 b | 30.33 ± 2.20 e | 2.77 ± 0.24 c | 9.59 ± 0.69 a |
| | CF | 24.76 ± 0.82 d | 2.64 ± 0.06 c | 29.35 ± 6.54 c | 43.18 ± 1.43 d | 4.60 ± 0.11 c | 6.86 ± 3.46 a |
| | Treatment | ** | * | * | ** | * | NS |
| | Year | NS | NS | NS | NS | NS | NS |
| | Treatment×Year | NS | NS | NS | NS | NS | NS |

Note: Values are the means ± standard errors for three biological replicates. Different letters in the column represent significant differences ($p < 0.05$) based on Duncan's test. * and ** represent $p < 0.05$ and $p < 0.01$, respectively, and NS represents "not significant". Abbreviations: nitrogen partial factor productivity (NPEP), nitrogen agronomic efficiency (NAE), nitrogen use efficiency (NUE), phosphorus partial factor productivity (PPEP), phosphorus agronomic efficiency (PAE), phosphorus use efficiency (PUE).

*3.6. Correlation Analysis*

The absorption of N was extremely positively correlated with P, K, Mg, Mn and Fe and was negatively correlated with Ca and S (Figure 11). The absorption of P was extremely positively correlated with K, extremely negatively correlated with Ca and was extremely positively correlated with Mn and Fe. The absorption of K was significantly or extremely negatively correlated with that of Ca and S, significantly or extremely positively correlated with that of Mn and Fe, and extremely negatively correlated with that of B. There was a significant or extremely negative correlation between Ca and Mg, Zn, Mn and Fe. However, there was a significant or extremely positive correlation between Mg and trace elements. S accumulation was positively correlated with that of Cu, Zn, Fe and B.

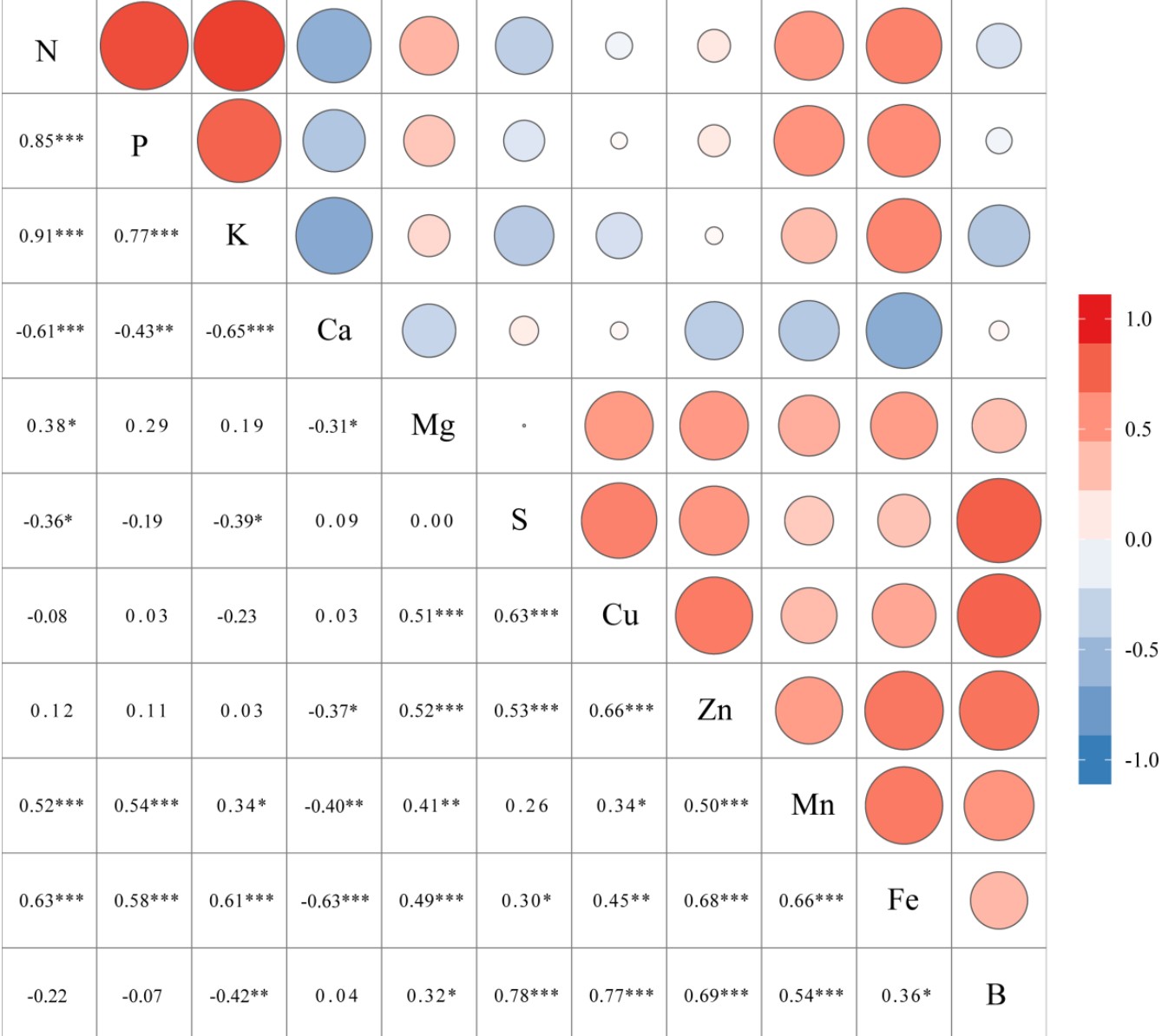

**Figure 11.** Correlation analysis of absorption of different elements. Note: * Statistical significance, i.e., $p < 0.05$; ** Statistical significance, i.e., $p < 0.01$; *** Statistical significance, i.e., $p < 0.001$.

Further path analysis shows (Figure 12) that Fe and Mn had direct effects on the absorption of N, P and K. Direct path coefficients were 0.591, 0.609, 0.732 and 0.783, 0.490 and 0.267 for Fe, Mn, N, P and K, respectively. B induced impacts on K absorption with the direct path coefficient of −0.660. A correlation among Mg and Zn and Fe absorption with path coefficients of 0.662 and 0.681. The correlation between S and Cu absorption and B absorption was noticed with path coefficients of 0.783 and 0.766.

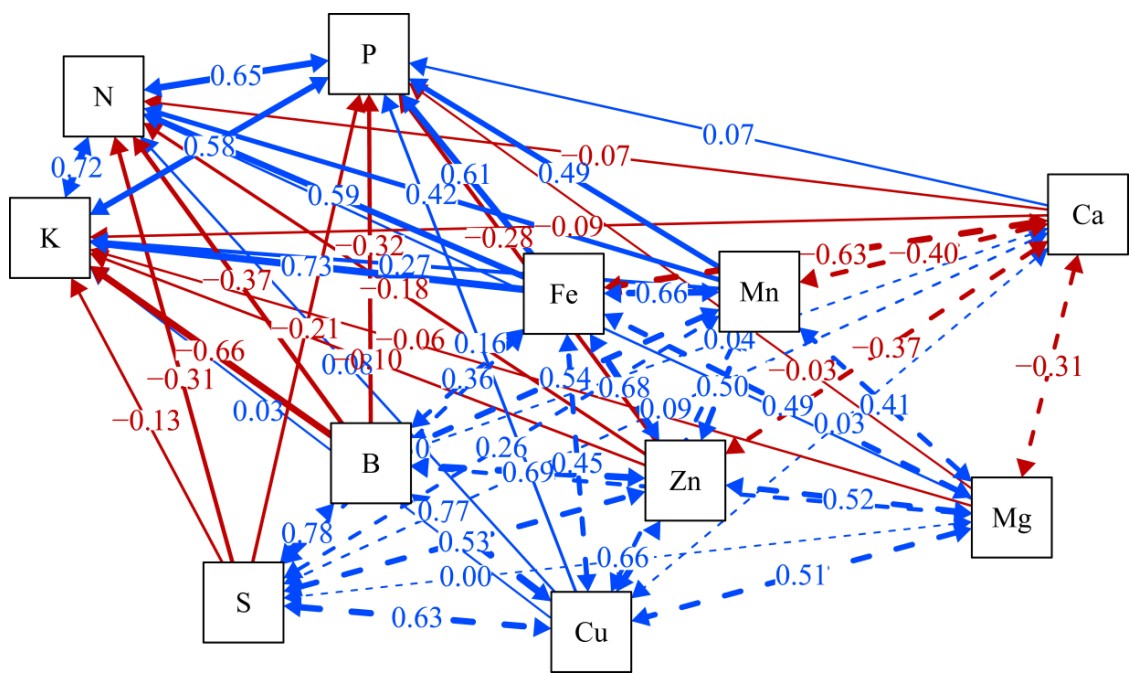

**Figure 12.** Path analysis of absorption of different elements.

## 4. Discussion

### 4.1. Effect of Organic Wastewater with Chemical Fertilizers on N, P and K Absorption

Organic plus inorganic fertilizer management can improve crop productivity, soil organic matter content and enhance the sustainability of different cropping systems [42]. In this study, application of organic wastewater and chemical fertilizers boosted soil Av-N, Av-K and Av-P and organic matter content compared with CF (Table 3). This shows that dripping organic wastewater can improve soil quality and soil fertility due to organic wastewater containing a large amount of small molecule soluble organic matter [43]. Another possible reason that application of organic wastewater with water drops in different times can provide a favorable environment for the soil microorganisms' growth [44], increases the availability of nutrients in the soil [31] and promotes the interaction and circulation of nutrients in the soil.

Compared with CF, the absorption of N, P and K under organic wastewater and chemical fertilizers increased significantly after 60 days of emergence, and showed an upward trend with the increased use of chemical fertilizers. It indicated that the application of organic wastewater and chemical fertilizers could reduce the use of chemical fertilizers by 20% which could still support the normal physiological activities of cotton. The reason may be that organic fertilizers have direct or indirect effects on soil nutrient content [45] and the nutrient absorption of crops [46]. Organic fertilizers can promote gene expression in roots and leaves which are related to the absorption and transport of nitrate nitrogen and ammonium phosphorus further improving nitrogen and phosphorus absorption [47].

Nutrient transportation and distribution in different organs is one of the important factors affecting the utilization efficiency of crops [48]. In this study, compared with CF, the distribution of N, P and K in reproductive organs was increased after the use of organic wastewater. It indicated that applying organic wastewater and chemical fertilizers could reduce the use of chemical fertilizers by improving the ability of nutrient absorption and transportation and organ competition [49]. In this study, F0.8 treatment had a higher N and K accumulation which was consistent with [16].

### 4.2. Synergistic Effect of Mn and Fe and the Increased Absorption of N, P and K

In the modern agricultural system, trace elements are crucial factors that limit plant growth and yield [50]. However, soil water and trace element deficiencies are common

in arid and semi-arid regions [51]. Foliar spraying of trace elements can significantly improve crop yield and quality [52,53]. In addition, organic wastewater can support crop growth because of its high content of trace elements such as Cu, Zn, Fe and Mg, which improves the crop chlorophyll content and photosynthetic rate [31]. Application of low concentration trace elements on the leaf surface or soil can induce and activate antioxidant enzymes, non-oxidative metabolism and sugar metabolism, thus alleviating the damage of oxidative stress [54]. In this study, compared with CF, organic wastewater and chemical fertilizers significantly increased the distribution of Mg, Mn and Fe in leaves and roots, and the accumulation of B and Zn in reproductive organs. However, after the dripping of organic wastewater, more use of fertilizers leads to less absorption of Zn, Fe, B and other elements, which may be due to the decrease in trace element accumulation caused by the yield dilution effect of high nitrogen fertilizer application [50].

The interaction between macronutrients and micronutrients affects plant growth and ion homeostasis [55]. Trace elements such as Cu, Mn, Fe and Zn are closely related to the metabolism of plants [56]. Mg can improve nutrient absorption and plant growth [57]. Under water stress, spraying Mg and K on leaves significantly increases the seed cotton yield and water use efficiency of cotton [58]. In this study, absorption of Mn and Fe was positively correlated with N, P and K, and path analysis further verified that the absorption of Fe and Mn by cotton was directly correlated with N, P and K. It indicates that dripping organic wastewater could promote the accumulation of N, P and K by increasing the absorption of Mn in cotton leaves and roots.

*4.3. Mechanisms of Cotton Yield and Fertilizer Utilization Efficiency*

Optimizing the fertilization strategy can improve crop productivity and reduce resource waste and environmental pollution [59]. The combined application of organic and inorganic fertilizers is an important measure to reduce the use of chemical fertilizers [20]. The use of organic liquid fertilizers can improve crop growth and productivity [60]. In this study, cotton yield was increased under dripping of organic wastewater compared with CF. The reason may be that dripping organic wastewater in different stages of the whole growth period [32] increased the content of available nutrients in the soil, improved the nutrient absorption environment of crops [31] and increased the absorption of N, Mn, Zn and other elements, thus enhancing the chlorophyll content and the photosynthetic rate [61]. However, after the dripping of organic wastewater, the cotton yield did not increase with the use of chemical fertilizers. This might be associated with the excessive nutrient supply resulting in vigorous growth of cotton and impairing the transport of assimilates to the reproductive organs [62]. In this study, cotton yield peaked under organic wastewater of 1329 kg ha-1 and the chemical fertilizer (N-$P_2O_5$-$K_2O$) of 182-104-76 kg ha$^{-1}$ application. Compared with the research of Wang et al. (2018), the yield level of this study increased by 16.3–19.0% under the same application amounts of fertilizers [63].

Combined application of organic fertilizers and chemical fertilizers increases partial productivity of nitrogen and phosphorus [64]. Application of industrial organic wastewater not only improves the utilization rate of nitrogen fertilizer, but also saves labor, fertilizers, and reduces environmental damage [32]. In the current study, partial productivity, agronomy utilization efficiency and apparent utilization efficiency of fertilizers was increased under the application of organic wastewater over CF. The reason may be that perennial excessive fertilization and straw returning resulted in the high content of fixed nutrients in the soil [65,66] and the application of organic wastewater accelerated the release of nutrients by several times [31]. However, adopting this fertilization mode for a long time, which reduces the use of chemical fertilizers, may speed up the consumption of soil nutrients. Climate, soil and management conditions should also be considered for the formulation of fertilization strategy [67]. The increase in fertilizer utilization efficiency was closely related to the high accumulation level of nitrogen and phosphorus maintained in the late growth stage [68]. In this study, the absorption of Mn and Fe by cotton enhanced the accumulation of N, P and K by cotton at the later growth stage.

## 5. Conclusions

Under a film drip irrigation system, a multiple time application of organic wastewater significantly increased cotton yield and the fertilizer utilization rate under F0.8 treatment. This was mainly due to the significant increase in the absorption of essential soil nutrients such as Mg, Cu, Zn, Mn, Fe and B. This further improved the absorption of N, P and K by plants and their distribution to reproductive organs. Correlation analysis showed a synergistic effect between the absorption of Fe and Mn and the absorption of N, P and K for cotton. Therefore, dripping organic wastewater of 1329 kg ha$^{-1}$ (organic mattered $\geq$ 20%, humic acid $\geq$ 20 g L$^{-1}$, *Bacillus subtilis* $\geq$ 2 $\times$ 10$^8$ L$^{-1}$) and application of chemical fertilizer (N-P$_2$O$_5$-K$_2$O) of 182-104-76 kg ha$^{-1}$ is a promising option to utilize industrial waste with reduced environmental pollution. These data provide new insights revealing that the use of chemicals combined with organic wastewater is the key to improve nutrient utilization efficiency by plants for improved yield with minimal environmental pollution.

**Author Contributions:** X.H.: conceptualization, investigation, formal analysis and writing of the manuscript; X.S.: methodology, formal analysis, investigation, data curation, writing—original draft; A.K. and N.L.: methodology, formal analysis; F.S.: resources, investigation; J.L. and Y.T.: methodology, investigation; P.H.: software, visualization; J.W.: writing—review and editing, supervision; H.L.: conceptualization, resources, supervision, writing review and editing, funding acquisition. All authors have read and agreed to the published version of the manuscript.

**Funding:** The authors are very thankful to the National Natural Science Foundation, China, which supported and provided funds for this project (32260539).

**Data Availability Statement:** The data presented in this study are available from the corresponding author upon request.

**Conflicts of Interest:** The authors declare no conflict of interest.

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
