# Peer review of "Industrial Organic Wastewater through Drip Irrigation to Reduce Chemical Fertilizer Input and Increase Use Efficiency by Promoting N and P Absorption of Cotton in Arid Areas"

_agriculture, doi:10.3390/agriculture12122007_

Round 1

Reviewer 1 Report

Research is relevant and interesting.

1. No research hypothesis is presented in the introduction;

2. I suggest checking the data of available phosphorus in the table No. 3;

3. In the figure No. 10  it is not explained what the different letters and vertical bars represent;

4. The quality of the figures  No. 1, 2 and 12  could be better.

Reviewer 2 Report

As a result of the evaluation of the article, it is recommended to make the following corrections.

Line 23:  ( Ammonium nitrogen): should be write lower case

Line 24: ( Available potassium): should be write lower case

Line 33: ( ha-1) should be corrected

Figure 8 and Figure 9: There is no need to use a percent sign (%) inside the figure, the figure looks too complicated (%) should be removed from inside the figure

In the overall evaluation, the article can be accepted after minor revision.

There are typos in the article, my advice is that the author should read and correct it.

Reviewer 3 Report

Line 76, use full form at first use, “nitrogen, phosphorus, potassium, calcium and phosphorus”

Figure 2 quality very low

The decimal of value reported in all tables of the manuscript must be uniform as the value of the detection limit of the used method.

Line 159-160: use full form at first use,

Line 169 to 180: provide suitable reference for equations

Figure 4, 5, and 6, quality very low

Line 244 and 249 and 267: remove abbreviation, only use full form in the legend

Line 293 legend of table 5? Remove typo

Go through journal guideline prepare citation according to that,

Please correct all grammatical errors and typo.

Recheck all abbreviations
